# Identification of Suitable Animal Welfare Assessment Measures for Extensive Beef Systems in New Zealand

**Y. Baby Kaurivi \*, Richard Laven, Rebecca Hickson, Kevin Stafford and Tim Parkinson**

School of Veterinary Medicine, Massey University, Palmerston North 4471, New Zealand;
r.laven@massey.ac.nz (R.L.); r.hickson@massey.ac.nz (R.H.); k.stafford@massey.ac.nz (K.S.);
t.parkinson@massey.ac.nz (T.P.)
**\*** Correspondence: Y.Kaurivi@massey.ac.nz; Tel.: +64-6350-5328

**Abstract:** Farm animal welfare assessment protocols use different measures depending on production systems and the purpose of the assessment. There is no standardized validated animal welfare protocol for the assessment of beef cattle farms in New Zealand, despite the importance of beef exports to the country. The aim of this study was therefore to identify welfare measures that would be suitable for an animal welfare assessment protocol for use in extensive pasture-based cow–calf beef cattle systems in New Zealand. The proposed animal welfare assessment measures were selected from the Welfare Quality protocol and the rangeland-based UC Davis Cow–Calf Health and Handling assessment protocol. Measures that were deemed impractical and/or unsuitable were excluded from the protocol. After testing the applicability of selected measures at one farm, additional measures that were deemed to be practical to undertake in New Zealand were identified and incorporated into the protocol. The intention was to identify animal welfare indicators that were assessable in the yard during a single farm visit, a questionnaire guided interview, and a farm resource assessment visit that evaluated cattle health and management. Further testing of the 50 measures that were identified as being appropriate will be undertaken on commercial beef farms to develop a practicable welfare protocol for extensive pasture-based beef systems.

**Keywords:** welfare assessment; extensive beef; New Zealand

## 1. Introduction

High animal welfare standards are increasingly being requested by consumers of animal products [1], so farm assurance schemes that include animal welfare are being used in many systems. This led to the development of animal welfare assessment protocols for a range of farm animal species in a range of production systems [2]. Developing an appropriate and practicable animal welfare assessment protocol requires the identification and selection of suitable welfare measures for each production system. The chosen measures need to be relevant, repeatable, and comparable across farms with similar production systems [3,4]. Critically, not all measures (whether animal or input-based) are applicable to all production systems. For example, in the New Zealand dairy system, measures such as hock lesions and rising restrictions when cows are standing up in stalls (both of which are essential measures in welfare assessments of housed cattle) are generally irrelevant because New Zealand dairy cattle are not normally housed [5]. Similar issues apply in regard to assessing welfare on extensive cow–calf beef farms in New Zealand, as the focus of many welfare assessments (e.g., welfare quality) is on growing cattle managed in intensive beef production systems, which are completely different to extensive operations [6]. On the other hand, although the rangeland-based UC Davis Cow–Calf Health and Handling assessment is based on an extensive system [7], it does not entirely fit the pasture-based

system in New Zealand. For example, this protocol is limited to the observation of cows in the handling chute only, and had limited ability to measure the human–animal relationship [4,8].

Thus, the aim of the current study was to utilize the Welfare Quality and UC Davis assessment protocols as the basis for identifying animal welfare assessment measures. It was hoped that this would form the basis of a practicable and credible animal welfare assessment protocol for the extensively reared pasture-based beef cow–calf systems in New Zealand.

## 2. Materials and Methods

The aim of this study was to identify valid measures that could be practically assessed on pasture-based beef farms in New Zealand, using a process of expert opinion and review followed by pre-trialing on one farm. In order to identify suitable animal welfare assessment measures that would suit the extensive pasture-based cow–calf beef systems which predominate in New Zealand, the Welfare Quality [6] and UC Davis [7] protocols were scrutinized by the authors, and measures that were deemed not suitable for New Zealand were excluded.

The remaining measures were then pre-tested by the authors along with an experienced technician (expert opinion) on one New Zealand beef farm to identify which of these measures could be practically assessed while cows were being handled for other purposes. In addition, further parameters that were not included in either the Welfare Quality or UC Davis protocols, but that were deemed relevant and practicable on the basis of the farm evaluation, were also identified.

The pre-testing of the potential protocol was undertaken in November (spring) on one beef farm in the Manawatu region in the south of the North Island of New Zealand (latitude 40.4° S). The climate is temperate, with a monthly average temperature range of 8 to 18 °C and an average rainfall of 900 mm/year. The farm was a 500 Ha mixed sheep and beef farm with river flats and hill country. The principal beef breed was Angus, with some Hereford and dairy-beef crosses. The cows were observed in the yard during weighing in a chute and hoof confirmation scoring in a holding pen. Additionally, a questionnaire-guided interview and farm resource visit captured health and management of cattle over the previous 12 months.

## 3. Results

Unsuitable criteria and measures that were excluded as not fit for an animal welfare protocol for New Zealand beef farming systems, with reasons for their exclusion, are presented in Table 1.

Of the 60 cows presented for observation at the farm, 50 cows were assessed in the yard. In total, 50 indicators (Table 3) were assessable during the yard observation, questionnaire guided interview, and farm resource visit. The selected measures were shown to be feasible and practicable on this farm, with the whole process taking 3 h. The measures identified as suitable and the methods of assessment are shown in Table 2. Additional parameters that were not in the Welfare Quality [6] and UC Davis [7] protocols, but were deemed relevant and practicable from the farm evaluation, are shown in Table 3.

**Table 1.** Excluded welfare assessment criteria and measures (from Welfare Quality [6] and UC Davis [7] protocols) and reasons for exclusion as unsuitable for extensive pasture-based beef systems in New Zealand.

| Welfare Principles | Criteria | Measures | Reasons of Exclusion |
|---|---|---|---|
| Good feeding | | Space at water troughs, cleanliness of water troughs, number and size of water points | Water provision on New Zealand beef farms is typically plentiful and is sourced mainly from natural resources such as streams and rivers. In semi-intensive and intensive systems, the 'provision of water' at 'clean water points' might be an easy task to accomplish, but in extensive systems as shown by Franchi [9] and Hernandez [10] in the tropics, water may be obtained from natural resources and, hence, challenging to assess. In the Welfare Quality protocol, the measure of 'absence of prolonged thirst', is indicated by the 'number and size of water points', but these measures have proved challenging to assess in tropical extensive systems [10], and thus were rejected as not suitable for extensive pasture-based systems as well. This measure was therefore replaced by access to clean and safe water and distance to water supply |
| Good environment | Access to pasture, housing, rising restrictions, hazards in indoor environment | | Cattle in New Zealand are already extensively reared on pasture and, as in accordance with Laven [5] and Hernandez [11], the 'access to pasture' criterion was not relevant. Another significant change to the Welfare Quality protocol was the replacement of 'good housing' with 'good environment'. With an absence of infrastructure or indoor obstacles in extensive systems where cattle might collide, the criteria of 'good housing' and 'rising restrictions' were thus excluded. Similarly, the absence of 'hazards in the indoor environment' was maintained but changed to capture hazards in extensive environments. |
| Good health | | Bloat | Bloated rumens in beef cattle on pasture are rare, therefore this measure was regarded as not suitable as it is more associated with grain-fed cattle [8]. |
| | | Hock lesions | Hock lesion evaluation was rejected by Laven [5] as a measure of extensive based dairy cattle and there is even less likelihood that they will be an issue in permanently grazed beef cattle on pasture. |
| | | Tail docking | Tail docking is prohibited by law [12] and thus not practiced in New Zealand beef cattle production systems. |
| Appropriate behavior | Expression of other behavior | Idling behavior | 'Expression of other behavior' is a subjective measure and may vary with assessors and in this study, this measure was discarded, as in agreement with Laven [5] and Hernandez [11] as a feature only assessable in confined cattle. Idling behavior was also found by Laven [5] to be unsuitable to be included for pasture-based dairy cattle as cattle spend more time grazing than idling and, hence, it has been excluded for beef cattle in the same environment. |

**Table 2.** Identified suitable measures and methods of assessment.

| Method of Assessment | Measures Assessed |
|---|---|
| Observation from an elevated platform next to the race while cattle were in the race | Body condition score, rumen fill score, broken tails, short tail integument alterations (abrasions, swelling, hair loss), tail twist, striking/hitting cattle with moving aids, cleanliness of the animals (flank, hind, udder, dirty tail (fecal soiling), coughing, hampered respiration, nasal discharge, ocular discharge, diarrhea, hoof condition |
| Observation on exit from chute | Lameness, exit speed (running or walking), stumble or fall, mis-catch |
| Video behavioral observation of cattle in holding pens for 20 min before animals were put in the race as a back-up for visual observation | Agonistic behaviors (head butting, displacement, fighting, chasing), cohesive behaviors (social licking, horning), qualitative behavior assessment (fearful, frustrated, bored, agitated, irritable, uneasy, distressed, tense, uncomfortable, etc.) |
| Noise observation around the yard | Noise of handlers, noise of equipment /machinery, presence/noise of dogs at yards |
| Yard design assessment; if it allowed easy movement of cattle from holding pens into forcing pens, the race, the chute, and exiting. | Handling facility design |
| Questionnaire guided interview with the farm manager to assess health and management of cattle in the last 12 months. | Castration and disbudding procedures, ear-tagging, frequency of yarding cows/year, disease history, animal health checks frequency, accidents/misadventures, mortality, vaccination, reproductive conditions (abortions, dystocia, prolapse) |
| Cattle in paddocks observation | Avoidance distance, overall body condition and any signs of health problems, behavioral observations. |
| Farm observation | Access and type of water supply, distance to water points, availability of shade in paddocks, feed/pasture condition, absence of hazardous objects/terrain |

**Table 3.** All potential measures identified as suitable (selected measures amalgamated from Welfare Quality [6] and UC Davis [7] assessment protocols and additional measures fit for New Zealand extensive pasture-based beef cattle farms).

| Welfare Principles | | Welfare Criteria | Combined Measures from Welfare Quality and UC Davis Assessment Protocols | Additional Measures Fit for New Zealand Beef Farms |
|---|---|---|---|---|
| **Good feeding** | 1 | Absence of prolonged hunger | Body condition score | Rumen fill score |
| | 2 | Absence of prolonged thirst | Access to safe and clean water | Distance to water points |
| **Good environment** | 3 | Comfort around resting | Cleanliness of the animals (flank, hind, udder) | Dirty tail |
| | 4 | Thermal comfort | | Availability of shade in paddocks |
| | 5 | Ease of movement | | Absence of hazardous objects/terrain |
| **Good health** | 6 | Absence of injuries | Lameness, integument alterations (abrasions, swelling, hair loss) broken tail | Short tail |
| | 7 | Absence of disease | Coughing, hampered respiration, nasal discharge, ocular discharge, diarrhea, hoof condition, mortality, reproductive conditions (abortions, dystocia, prolapse), | Disease history, animal health checks frequency, accidental misadventures (accidents where cattle fall in tomos (sinkholes) or roll off hills) |
| | 8 | Absence of pain induced by management procedures | Disbudding/dehorning, castration, | Ear-tagging |
| **Appropriate behavior** | 9 | Expression of social behaviors | Agonistic behaviors, cohesive behaviors | |
| | 10 | Negative emotional state | Qualitative behavior assessment (fearful, frustrated, bored, agitated, irritable, uneasy, distressed, tense, uncomfortable, etc.) | Handling facility design, noise of handlers, noise of equipment/machinery, presence/noise of dogs at yards, frequency of yarding cows/year |
| | 11 | Good human–animal relationship (stockmanship) | Avoidance distance, handling measures (mis-catch, tail twist, striking/hitting cattle with moving aids, stumble, fall or run exiting | |

## 4. Discussion

### 4.1. Excluded Measures

It was evident during the study that not all measures (whether animal- or input-based) were applicable to the production system under study. The exclusion of measures (Table 1) was based on eliminating animal welfare criteria and measures that were not applicable to the extensive beef production system in New Zealand. The main criteria and measures that were rejected were largely those that had previously been rejected or questioned by Laven [5] as not practicable to apply for dairy cattle animal welfare assessment in extensive pasture grazing systems in the country. These measures were primarily those that were relevant to intensively-managed, housed cattle; including access to pasture, idling behavior, housing and rising restrictions. These measures were also irrelevant for assessment of beef cattle reared on extensive pasture.

Another consideration used to exclude certain measures such as cleanliness and 'space at water troughs' was based on the applicability of the Welfare Quality [6] protocol that has been verified under tropical conditions [9–11]. Moreover, [4,8], who also evaluated the UC Davis system in extensively-managed cattle, also rejected bloating.

### 4.2. Suitable Welfare Assessment Measures Identified from the Selected Protocols

The applicability of selected measures (Table 2) was based on the Welfare Quality [6] and UC Davis [7] protocols. A few adjustments were required to produce a practicable assessment on extensive commercial beef farms. For example, time limitation did not permit locomotion score or body condition score (BCS) beyond lame or not, and fat, good or thin, respectively.

### 4.3. Feeding

BCS has been regarded as the preferred indicator for good feeding in most on-farm welfare assessments [13]. The resource based measuring of 'absence from prolonged hunger' with indirect, non-validated measures such as 'centimeters of feeding trough per cow' or measuring the feed consumed by a group or individual cattle [13,14] has largely been replaced by measuring the BCS of the animals. In the present study it was more practical, with the limited time available per animal, to scale the BCS to good condition, fat or thin animals than to assess them based on the New Zealand beef cattle scale of 1 to 10 [15], or even the European/North American scale of 1 to 5.

Assessment of 'water availability' especially determining space, cleanliness, number, and size of water points is difficult to assess in extensive systems, so this was simplified to a simple yes/no regarding access to clean fresh water and the distance that cattle had to walk to access such water. This adjustment of distance is valuable in extensive systems, as the longer the distance that livestock travel to water the lower the rate of pasture utilization [16,17]. In extensive systems this definition is simpler and easier to use than the range of measures included in 'water availability' and is also supported by scientific evidence.

### 4.4. Environment

The key criteria for good environment were identified as 'thermal comfort' and 'ease of movement'. However, the concept of 'thermal comfort' was not developed nor assessed in the Welfare Quality [6] protocol, so this criterion was amended to 'comfort with access to shade or shelter', as was proposed by Hernandez [11]. Heat stress reduces the comfort and performance of cattle [18,19] which led Hernandez [11] to suggest replacing 'lying outside the lying area' with 'lying in the shade' as more suitable for application to extensive pastured cattle. However, this indicator is difficult to assess in the field, as beef cattle at pasture lie down mostly at night and are more likely to be found standing than lying when resting during daylight hours [20]. However, availability of shade for grazing cattle, for example from trees, shrubs, galleys, and man-made canopies, does have a significant effect

on animal productivity and well-being [21] and is thus likely to be a useful alternative to animal observation of lying behavior in extensive pasture-beef systems.

The assessment for 'comfort around environment' in the Welfare Quality protocol evaluates the cleanliness of the hind and lower legs, flank, and udder [6]. However, in the current study, the term was adjusted to 'dirtiness of cows' as it was deemed a better fit for this criterion. Laven and Fabian [5] included dirty flanks, hindlimbs, and udders in their assessment of the physical condition of pasture-based dairy cattle, but commented that the impact of dirtiness on welfare was likely to be different in pasture-based cattle and housing-based systems. This is consistent with the common finding in beef cattle in New Zealand, that beef cattle commonly have dirty lower limbs (particularly after wet weather) without significant consequences [22,23]. Therefore, in the present protocol only the dirtiness of the upper hind limbs was recorded.

### 4.5. Cattle Health

The relevant indicators for good health were selected as: 'absence of injuries and disease', 'management of painful procedures', and 'mortality'. To demonstrate absence of injuries and disease, animal-based measures were typically selected as described in the Welfare Quality [6] and UC Davis [7] protocols. For painful procedures, welfare assessment is influenced by countries' laws and societal expectations. In the European-based Welfare Quality protocol, data are collected about the method used for procedures such as castration, as well as the use of anesthetics and analgesics. However, in many countries which have extensive farming systems, (including New Zealand at the time of the on-farm examination: November 2017) painful procedures, including disbudding, could legally be performed without analgesics or anesthetics in young animals less than 9 months old [12]. However, it is clear that, irrespective of legislation, procedures such as castration and disbudding are painful, and that this pain can be mitigated by using anesthetics and analgesics [24]. Thus, in a welfare assessment it is important to assess pain management during and after painful procedures, even if pain management is not required by legislation. Welfare assessment should reflect our understanding of animal welfare science and not be controlled by it. For example, the regulations in New Zealand have recently changed, so that as of October 2019 disbudding without local anesthetic will be prohibited.

The Welfare Quality protocol assesses overall mortality, only recording the percentage of animals which died whatever the cause (for example whether it was disease or accident and whether the animal died or was euthanized) [6]. However, in New Zealand the hazards posed by the topography of hill farms and rolling country mean beef cattle are prone to misadventure. Thus, instead of collectively assessing overall mortality, cases of misadventure deaths on the farm during the last 12 months were separated from death due to disease or euthanasia for health issues.

### 4.6. Cattle Behavior in the Yard

Hernandez [11] suggested that the 'expression of social agonistic behavior' which includes head-butting, displacement, chasing, and fighting should be observed remotely (i.e., using binoculars) so as not to disturb pasture-based cattle. However, social agonistic behavior probably accounts for <5% of the time budget of grazing beef cattle [20]. Thus, in order to observe sufficient agonistic behavior at pasture to make a reasonable assessment of behavior, cattle would have to be observed for a prolonged period. It was decided to assess social behavior in the yard where agonistic behavior is likely to be more frequent [25].

'Positive emotional behaviors' were initially combined as 'calm/content' in this study as was recommended by Hernandez [11], even though these authors were unconvinced that the assessment of positive behaviors would be worthwhile (i.e., that such observations might be both practicable and of limited value). In the present study, it was difficult to quantify and assess these behaviors during the welfare assessment at the one farm. This criterion was rejected in favor of only focusing on assessing negative behaviors.

It was still difficult to follow the Welfare Quality protocol in order to quantify and assess 'negative qualitative behaviors' (fearful, frustrated, bored, agitated, irritable, uneasy, distressed, tense, uncomfortable, etc.) [6], during the farm visit. All of these behaviors can only be deduced subjectively from the body language of cattle [26] and are difficult to assess accurately. Thus, 'negative emotional behaviors' were merged into 'negative behavior', as a way of recording to describe general negative body language and behavior. For example, negative behavior (in the holding pens) could describe those cattle with tails swishing in agitation, shuffling feet, pawing, running away, and bellowing, pacing, head-shaking, and kicking. Each individual action was rare, so the data were amalgamated to create a measure which was the total count of negative behaviors as a percentage of observed cattle.

### 4.7. Appropriate Stockmanship

Cattle handling measures that indicate stockmanship in and out of the chute (vocalizing, stumbling, balking, falling or, running) were according to their description in the UC Davis [7] protocol. 'Tail twisting' and 'mis-catching' of cattle in the race and chute, respectively, were also identified as practicable measures. However, assessment of the 'use of moving aids' was challenging, as the definition was not clear. Simon [4] also disregarded 'moving aids touches' as a measure in an ideal comparable welfare measuring protocol because of its unclear definition. For example, some handlers would gently touch cattle with a moving aid as an extension of their arm while others used it more aggressively and on sensitive points. Thus, in the present study, hitting cows with moving aids was adjusted to evaluate the collective percentage of cows hit or poked regardless of the position on animals that were hit.

Measuring avoidance behavior was changed from the method used in the Welfare Quality [6] protocol because unlike dairy cattle and intensively reared beef cattle that are in daily contact with humans; extensive beef cattle tend to have a longer flight distance [25,27,28]. Therefore, this individual measure was replaced by the withdrawal distance when approaching a herd of cows in a paddock. However, although measurement of this distance is feasible, its interpretation in extensively-reared beef cattle may be problematic.

### 4.8. Additional Measures Identified as Suitable for New Zealand Extensive Pasture-Based Cow–Calf Beef Farming Systems

The extra measures that were deemed necessary for New Zealand pasture-based beef systems were based on the peculiarities of this system. For example, the topography of New Zealand hills and high country pose the risk of cattle falling downhill. Additionally, cattle may fall into tomos (natural underground holes), gullies, and bogs in some areas of New Zealand [23]. The 'ease of movement' was thus assessed by considering the absence of hazardous objects or risky areas.

Another peculiar practice of extensive beef farming where cattle are predominantly raised on hills in New Zealand is herding cattle with dogs and motorbikes. Herding cattle into the yards with other animal species like dogs or horses may cause them distress and thus, as observed by Hernandez [11], such farms should be recorded. Similarly, herding cattle on motorbikes can cause fearful behavior in cattle and thus both measures were noted as potentially useful indicators of extensive beef cattle welfare.

Sudden exposure to loud noise can stress cattle and may result in a stampede and other fear reactions [27]. Therefore, the noise made by handlers, equipment, and dogs during cattle handling were considered as potential welfare assessment measures. This opinion is supported in the measures of beef cattle welfare indicators developed by the World Organisation for Animal Health (OIE) that indicated that, extensively-managed cattle should be herded and handled calmly with minimal noise and at a slow pace to avoid stress [29].

The herding of cattle into the yards is known to influence the behavior of cattle in the yard during handling and restraining [25]. Thus, for extensively-managed cattle that are rarely yarded and handled, the 'yarding frequency per year' might serve as an indicator of welfare on beef farms. Moreover, the

number of times farmers inspect extensive-based cattle is crucial to their health. Therefore, 'health check frequency', particularly during the calving season, was identified as a valuable potential indicator for extensive beef farms welfare assessment. In New Zealand, the welfare concern for the traditional weaner production and finishing beef production farming is mainly due to limited inspection of cows and calves, and thus dystocia cases and injured or sick cattle might not receive immediate attention [23,30,31].

The design of yards and handling infrastructure can easily affect the flow and behavior of cattle in the yard [25,32]. 'Facility design' was added to evaluate any negative impact it might have on animal handling and flow in the holding pens and the race. Another added measure was, 'agitation and/or fearful behavior' of individual cattle in the race such as climbing on other cattle or attempting to escape. Such behaviors were recorded as indicators of negative behavior in the chute and as a measure of stockmanship.

Dirtiness in the Welfare Quality protocol focuses on dirt on the udder, hindlimbs, and flanks, but not the tail. In New Zealand, it is not unusual to see a ring of hardened feces on the tail of cattle. This ring can constrict the circulation to the distal part of the tail and, if it persists for long enough, it can result in necrosis and sloughing off and shortening of the tail predominantly in beef cattle. The risk factors for this condition are poorly understood, but it is thought to be associated with the production of large volumes of soft feces (typical of cattle on improved pasture). Thus, 'dirty tails' and 'short tails' were added to the list of potential measures to be recorded as measures of good environment and health respectively.

The 'disease history' measure was also added to the questionnaire for the New Zealand protocol to capture health and welfare of beef cattle in the last 12 months. This was deemed a more comprehensive approach than just performing a physical evaluation of the cows. Despite disease status of beef cattle in New Zealand being generally good, at least partly because of the absence of major diseases [23], the health and welfare status of an individual beef farm will be significantly influenced by the prevalence of diseases, such as bovine viral diarrhoea (BVD), facial eczema and milk fever, and reproductive problems like dystocia.

'Ear tagging' was not included in the original Welfare Quality protocol. However, it does cause pain and can lead to complications such as ear infection. Hernandez et al. [11] suggested inclusion of ear tagging in a cattle welfare assessment, but their study did not indicate a suitable measure for assessment. One potential method of assessment could be to determine the age at which tagging takes place. Some farmers ear-tag calves at a later stage, potentially causing more pain than if placed in younger animals. Younger animals are also easier to handle and quicker to recover than older animals [24].

In housed cattle, feed supply and access are crucial measures of animal welfare. In the extensive pasture system, these are more difficult to assess. A potential alternative is the scoring of rumen fill, which responds more quickly to insufficient feed than body condition score, as there are individual differences in rumen fill between cows (indeed these were observed on the trial farm), rumen fill provides an individualized measure of short-term feed intake. It has been commonly used as an indicator of the nutritional status of cattle [33,34], though commonly in dairy rather than beef cattle. To facilitate scoring the 5 point 'rumen fill score' was simplified to empty, normal, and distended.

## 5. Conclusions

The identification of suitable animal welfare measures is crucial for any valid and practicable welfare assessment across different production systems as all measures are not necessarily applicable to all systems and regions. The purpose of the current study was to identify suitable animal welfare assessment measures for extensive cow–calf beef systems in New Zealand. It was important to select valid measures with limited subjectivity that can be practically assessed across beef farms. Additional measures, such as presence of short tail in cows, deaths due to misadventure, yarding frequency, and herding of cattle with dogs and bikes were also identified as potentially valuable measures for

the assessment of extensive beef farms. Although the current study was based on a single farm, the 50 measures that were identified are anticipated to be relevant to other beef farms in the country. Thus, the next step is to test the practicability of applying these measures on representative commercial beef farms with the aim of paving the way to a validated animal welfare assessment protocol for extensive beef farming enterprises such as those in New Zealand.

**Author Contributions:** Y.B.K.—Main researcher who did the project proposal and data collection, analysis, and main write up; R.L.—Main supervisor who did conceptualization, methodology, funding acquisition, review, and editing; R.H.—conceptualization, formal analysis; K.S.—conceptualization, methodology and editing, and validation; T.P.—review and editing.

**Funding:** This research received no external funding, but received funding from Massey University SoVS.

**Acknowledgments:** Dean Burnham for technical support and initial assistance at the farms.

**Conflicts of Interest:** The authors declare no conflict of interest.

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
