# Peer review of "Identification of Suitable Animal Welfare Assessment Measures for Extensive Beef Systems in New Zealand"

_agriculture, doi:10.3390/agriculture9030066_

Round 1

Reviewer 1 Report

Line 35-36: what are “rising restrictions”?

What were the criteria for determining whether a parameter was suitable or not?

Table 1: While I appreciate the challenges surrounding evaluation of water access in an extensive system, complete omission of water evaluation in a welfare assessment is not acceptable.  Husbandry non-negotiables include food, water, and shelter – so I would recommend that the authors revise to a yes/no parameter.  Do the cattle have access to a safe and clean water source? Yes or No.  To not report anything on water availability will leave the final assessment remiss of a critical component of welfare. 

I can now see that distance to water source is included in Table 3, however, there is no method to report whether drinking water is actually available.  What is the likelihood that water is available during dry periods?

I am ignorant regarding the seasonality of the environment that these animals are housed upon, but ranging cattle can become bloated if they are grazed on lush grass after a period of drought or access to poor forage quality.  Would there be seasonal concerns for bloat as the pasture conditions change?

Table 2: I can see some challenges associated with including a video behavioral observation component in the welfare assessment scheme.  The video recordings would need to be decoded by the auditor to give the producer useful data – and this seems unrealistic to implement in an on-farm audit scenario.

Table 2: Noise observations around the yard: this seems subjective.  Are the evaluators simply assessing a yes/no or are there criteria/thresholds that noise levels are compared against?

Table 2: Language around “questionnaire guided…” is different compared to the other methods of assessment described.

Table 3: do the authors mean “rumen fill score”?  it is my understanding that rumens can only be evaluated at the slaughterhouse.

Table 3: “misadventures”?  while that sounds like fun, I think it would be difficult to quantify.

Line 62: typo

Lines 120-125:  oh my!  Sounds like an adventure to Mordor!

Line 123: Figure 1 is mentioned in the text, but there is no figure included in the manuscript.

Line 156: Figure 2 is mentioned in the text, but there is no figure included in the manuscript.

How did the researchers numerically evaluate the revised and new metrics for assessment?  The WQ has a numerical scoring and calculation component for reporting and threshold evaluation.  There is no mention of how these new metrics/metric suite will be quantified in a way to give producers the needed feedback.

Are they evaluating all animals on farm?  How are the observations scored?  What information is provided back to the producer? 

It seems as though these new “metrics” need to be evaluated on more than one operation before they can be deemed as suitable for evaluating welfare on farms across New Zealand.  Also – if the climate varies substantially/seasonally throughout New Zealand, then that could influence the usefulness/impactfulness of the variables.

Author Response

Line 35-36: what are “rising restrictions”?

Rising restrictions of cattle standing or moving that are common in stalls in intensive production systems. I have added “in stalls’ in line 36 just to make it clearer.

What were the criteria for determining whether a parameter was suitable or not?

The measures needed to be relevant to the production system, practicable and less subjective.

As stated in 4.1 “The exclusion of measures (Table 1) was based on eliminating animal welfare criteria and measures that were not applicable to the extensive beef production system in New Zealand. The main criteria and measures that were rejected were largely those that had previously been rejected or questioned by [5] as not practicable to apply for dairy cattle animal welfare assessment in extensive pasture grazing system in the country. These measures were also irrelevant for assessment of beef cattle reared on extensive pasture”

Also as stated in Introduction line 32-33; ‘The chosen measures need to be repeatable and comparable across farms with similar production systems’. I have added “relevant’ in line 32 to be clearer.

Also in line 50-51 “to identify valid measures that could be practically assessed on pasture-based beef farms in New Zealand” determined whether the parameters were suitable or not.

Table 1: While I appreciate the challenges surrounding evaluation of water access in an extensive system, complete omission of water evaluation in a welfare assessment is not acceptable.  Husbandry non-negotiables include food, water, and shelter – so I would recommend that the authors revise to a yes/no parameter.  Do the cattle have access to a safe and clean water source? Yes or No.  To not report anything on water availability will leave the final assessment remiss of a critical component of welfare. 

I can now see that distance to water source is included in Table 3, however, there is no method to report whether drinking water is actually available.  What is the likelihood that water is available during dry periods?

Noted. Access to a clean safe water supply is implicit in the calculation of the distance to the water supply, but we agree with reviewer that we should have made it explicit. We have now done so in both table 1 and table 3

I am ignorant regarding the seasonality of the environment that these animals are housed upon, but ranging cattle can become bloated if they are grazed on lush grass after a period of drought or access to poor forage quality.  Would there be seasonal concerns for bloat as the pasture conditions change?

In dairy cattle in New Zealand there is a significant seasonal bloat risk. However, this is related to high quality high clover pastures. Beef cattle are generally kept on poorer quality pasture as well as eating significantly less. As such bloat is not recognized as a significant risk on New Zealand beef farms. .

Table 2: I can see some challenges associated with including a video behavioral observation component in the welfare assessment scheme.  The video recordings would need to be decoded by the auditor to give the producer useful data – and this seems unrealistic to implement in an on-farm audit scenario.

Video will reduce numbers of assessors and also gives back-up to the assessment of on-farm cattle

Table 2: Noise observations around the yard: this seems subjective.  Are the evaluators simply assessing a yes/no or are there criteria/thresholds that noise levels are compared against?

Yes, but although it is a subjective measure, it can be classified: good welfare- no noise, sufficient welfare- minor noise and for poor welfare -noisy handlers, dogs and machinery.

Table 2: Language around “questionnaire guided…” is different compared to the other methods of assessment described.

Noted. Language adjusted accordingly.

Table 3: do the authors mean “rumen fill score”?  it is my understanding that rumens can only be evaluated at the slaughterhouse.

Noted and adjusted to rumen fill score

Table 3: “misadventures”?  while that sounds like fun, I think it would be difficult to quantify.

Noted and adjusted to adjusted “accidental misadventures”. Meant as accidents (when cattle fall in tomos or roll off hills) where the number of cases can be quantified.

Line 62: typo

- [KB1] 

Lines 120-125:  oh my!  Sounds like an adventure to Mordor!

J

Line 123: Figure 1 is mentioned in the text, but there is no figure included in the manuscript.

Noted and figure removed.

Line 156: Figure 2 is mentioned in the text, but there is no figure included in the manuscript.

Noted and figure removed.

How did the researchers numerically evaluate the revised and new metrics for assessment?  The WQ has a numerical scoring and calculation component for reporting and threshold evaluation.  There is no mention of how these new metrics/metric suite will be quantified in a way to give producers the needed feedback.

The calculation of welfare scores was beyond the scope of this paper. The aim of the current study was just to identify measures that were suitable for assessment. This was a one-farm study to assess feasibility of this assessment. When we take this protocol on to commercial farms we will look at scor.

Are they evaluating all animals on farm?  How are the observations scored?  What information is provided back to the producer? 

 From the results line 72- “Of the 60 cows presented for observation at the farm, 50 cows were assessed in the yard”. The target was to assess more than 50% of cows in a herd on a convenience basis.

This was a preliminary study to identify measures that can be practically assessed on one pasture-based farm in New Zealand and thus, therefore, statistical data analysis was not required.

As discussed for scoring, as a one farm assessment there was no intention to feedback to the farmer (and it was a Massey University farm). For future projects presentation of the data will be an important part

It seems as though these new “metrics” need to be evaluated on more than one operation before they can be deemed as suitable for evaluating welfare on farms across New Zealand.  Also – if the climate varies substantially/seasonally throughout New Zealand, then that could influence the usefulness/impactfulness of the variables.

Yes, the 50 measures that were identified are anticipated to be relevant to other beef farms in the country. Thus, the next step is to test the practicability of applying these measures on representative commercial beef farms with the aim of paving the way to a validated animal welfare assessment protocol for extensive beef farming enterprises such as those in New Zealand. That is what we are currently doing

Yes, seasonality could influence the welfare outcome. Thus, for the commercial farms we will assess them during pregnancy testing (summer) at the animal level and in winter at the herd level .  

Reviewer 2 Report

The reviewed manuscript deals with the development of a suitable animal welfare assessment for free-ranging cattle in New Zealand. The topic is up-to-date and of interest to the scientific community.

In general, the manuscript is written accurately and easy to follow and of adequate quality.

With regard to the content, I only miss an outlook how the identified measures could be transferred to other countries or if they are restricted to New Zealand, e.g. what other countries could learn from this study (now or later).

Another minor point is the introduction: This demands quite a lot of previous knowledge from the reader. Of course you could expect it, but it might be helpful to explain a little on the background of the two modified assessment protocols. For sure you should cite them when first mentioning (line 39,41), not only in the Material and Methods part. I would cite them each time and also in the Tables, when referred to both sources.

From a formal point of view, I recommend to cite the authors and not only the respective numbers, in case the sentence is built as this, for instance: …as shown by (9) and (10) (better: as shown by Meyer (9) and Müller (10))

This is relevant for the whole text, and also for the tables. Please correct.

In Table 1, column 1, is housing crossed out by purpose?

Table 3: What are the 50 potential measures mentioned in the title? Everything in the table?

Author Response

Reviewer 2

The reviewed manuscript deals with the development of a suitable animal welfare assessment for free-ranging cattle in New Zealand. The topic is up-to-date and of interest to the scientific community.

In general, the manuscript is written accurately and easy to follow and of adequate quality.

With regard to the content, I only miss an outlook how the identified measures could be transferred to other countries or if they are restricted to New Zealand, e.g. what other countries could learn from this study (now or later).

As concluded ‘The current study was based on a single farm where the 50 measures that were identified are anticipated to be relevant to other beef farms in the country. Thus, the next step is to test the practicability of applying these measures on representative commercial beef farms with the aim of paving the way to a validated animal welfare assessment protocol for extensive beef farming enterprises such as those in New Zealand”

Thus, for now the measures identified were those deemed practicable for New Zealand beef farms exclusively. Meaning that after testing the measures at commercial farms, some measures might be found not relevant or practicable.  Then measures that are relevant for other production systems such as those in Namibia semi-desert rangeland can be adjusted accordingly.

What other countries can learn from this study is that not all measures are applicable to all production systems and simply transferring protocols is not feasible.

Another minor point is the introduction: This demands quite a lot of previous knowledge from the reader. Of course you could expect it, but it might be helpful to explain a little on the background of the two modified assessment protocols. For sure you should cite them when first mentioning (line 39,41), not only in the Material and Methods part. I would cite them each time and also in the Tables, when referred to both sources.

Good suggestion. Noted and cited accordingly in line 40, 42, Table 1 and Table 3 headings, line 13, 17 and 18, 36, 45, 57, 58, 73, 91, 103, 112

From a formal point of view, I recommend to cite the authors and not only the respective numbers, in case the sentence is built as this, for instance: …as shown by (9) and (10) (better: as shown by Meyer (9) and Müller (10))

This is relevant for the whole text, and also for the tables. Please correct.

Noted and corrected in the document

In Table 1, column 1, is housing crossed out by purpose?

Yes, housing was crossed out because it was replaced by “environment” as explained in the reasons for exclusion.

Table 3: What are the 50 potential measures mentioned in the title? Everything in the table?

Yes, all 50 measures in the table. But adjusted title to remove the number 50 

Round 2

Reviewer 1 Report

Thank you for the revised version of your manuscript.  Several items are much more clear to the reader. 

However, there are a few points that need to be addressed.

-          Most of the readership will not have been to New Zealand, nor are familiar with the local vernacular, so please ensure that you minimize jargon and explain things sufficiently so a reader that has never been to this geographical location can understand what you’re referring to.

-          Specific examples:

o   Still unclear about what “rising restrictions” are.  This may just be an issue of language or jargon.  For rising restrictions, are you referring to restrictions on how high up an animal can stand?  The criteria in which they are raised from calf to adult?  This is not a common term in North America, so it is confusing to identify to what you are specifically referring.

o   For “misadventures” please explicitly state that these are defined as “accidents where cattle fall in tomos (sinkholes) or roll off hills” 

-          Video decoding is very labor intensive.  On-farm welfare audits can take upwards of 9 hours to conduct.  Therefore, adding a video decoding component to an on-farm audit 1) increases the workload of the audit and will therefore not likely be conducted and 2) delays the feedback to the producer because they have to wait for the video to be decoded.  Yes, it’s only at 20 minute clip, but it can take upwards of an hour to decode a 20-minute video clip – especially for those that are not experienced in video decoding and for the metrics that you are proposing to include in the evaluation.  Auditors would need to be trained and validated, and an ethogram developed to make this a reliable and valid metric.  Perhaps suggest video recordings as a backup to visual observations, but would recommend this not be the sole method of information collection for the audit itself.

-          Cattle respond differently to different types of noise (human vs. metal).  Suggest adding type of noise to the evaluation and then evaluating good-sufficient-poor within that framework.

Author Response

However, there are a few points that need to be addressed.

-          Most of the readership will not have been to New Zealand, nor are familiar with the local vernacular, so please ensure that you minimize jargon and explain things sufficiently so a reader that has never been to this geographical location can understand what you’re referring to.

-          Specific examples:

o   Still unclear about what “rising restrictions” are.  This may just be an issue of language or jargon.  For rising restrictions, are you referring to restrictions on how high up an animal can stand?  The criteria in which they are raised from calf to adult?  This is not a common term in North America, so it is confusing to identify to what you are specifically referring.

Yes, it is rising restriction for how an animal finding it difficult to stand up or collide into obstacles due to the restriction of the size (height and space) of the stall.  It is the same used in the Welfare QualityR) protocol. It is for dairy cows raised in stalls. But I understand why you find it unclear and have explained clearer in line 34.

·         For “misadventures” please explicitly state that these are defined as “accidents where cattle fall in tomos (sinkholes) or roll off hills”

Noted. Added “accidents where cattle fall in tomos (sinkholes) or roll off hills” after accidental misadventures in Table 3 under welfare criteria 7

-          Video decoding is very labor intensive.  On-farm welfare audits can take upwards of 9 hours to conduct.  Therefore, adding a video decoding component to an on-farm audit 1) increases the workload of the audit and will therefore not likely be conducted and 2) delays the feedback to the producer because they have to wait for the video to be decoded.  Yes, it’s only at 20 minute clip, but it can take upwards of an hour to decode a 20-minute video clip – especially for those that are not experienced in video decoding and for the metrics that you are proposing to include in the evaluation.  Auditors would need to be trained and validated, and an ethogram developed to make this a reliable and valid metric.  Perhaps suggest video recordings as a backup to visual observations, but would recommend this not be the sole method of information collection for the audit itself.

Agreed. Video decoding was indeed a laborious exercise and took your suggestion to be a backup for visual observations in Table 2:

 “Video behavioural observation of cattle in holding pens for 20 minutes before animals were put in the race as a back-up for visual observation”

-          Cattle respond differently to different types of noise (human vs. metal).  Suggest adding type of noise to the evaluation and then evaluating good-sufficient-poor within that framework.

Noted:  To be added to the measures done at commercial farms for the next article. However, the type of noise was separated already in Table 3 under Welfare Criteria 11 as follows:

“ noise of handlers, noise of equipment /machinery, presence/noise of dogs at yards”

Reviewer 2 Report

The manuscript has been improved according to my comments and the open questions were answered. Therefore, I recommend its publication in Agriculture.

Author Response

Thank you for your kind acceptance of our paper. No comments were required.